# Masked Diffusion as Self-supervised Representation Learner

## Abstract

Denoising diffusion probabilistic models have recently demonstrated state-of-the-art generative performance and have been used as strong pixel-level representation learners. This paper decomposes the interrelation between the generative capability and representation learning ability inherent in diffusion models. We present the masked diffusion model (MDM), a scalable self-supervised representation learner for semantic segmentation, substituting the conventional additive Gaussian noise of traditional diffusion with a masking mechanism. Our proposed approach convincingly surpasses prior benchmarks, demonstrating remarkable advancements in both medical and natural image semantic segmentation tasks, particularly in few-shot scenarios.

## 1 Introduction

Diffusion models (Sohl-Dickstein et al., 2015; Ho et al., 2020) have showcased impressive image synthesis abilities by iteratively generating cleaner images from noisy ones using a Gaussian distribution. Recently, novel methods such as cold diffusion (Bansal et al., 2022) have aimed to substitute diffusion's denoising step with other processes. Unfortunately, such efforts risk deviating from the theoretical underpinnings of diffusion (see Section 3.1), which heavily rely on the Gaussian distribution and the introduction of Gaussian noise into images. Furthermore, the unsatisfactory results of cold diffusion seem to underscore the indispensable nature of denoising in the diffusion process.

Fortunately, denoising's significance diminishes when one focuses on the self-supervised pre-training facet of diffusion (e.g., Baranchuk et al. (2022)), which employs intermediate activations from a trained diffusion model for downstream segmentation tasks. As emphasized in Section 3.1, training a denoising diffusion model can be simplified by reconstructing the original image (or its added noise equivalent) from a noisy version, with the noise being modulated by various timesteps. Thus, viewed as a representation learning technique, diffusion can be seen as an autoencoder performing denoising across various complexities. In this context, the denoising operation loses its centrality, as evidence is inconclusive that denoising pre-training inherently produces better representations compared to recovering from alternative corruptions (e.g., masking).

Moreover, we identify a mismatch between the pre-training generative task and the downstream dense prediction task, leading to performance degradation during fine-tuning. Namely, high-level, low-frequency structural aspects of images are demanded for a dense prediction task (e.g., semantic segmentation) in few-shot scenarios, where the trainable network is designed to be lightweight to avoid over-fitting. Contrastingly, generative models often allocate a significant portion of their capacity to capture low-level, high-frequency details, as highlighted in Ramesh et al. (2021).

Motivated by these insights, our study diverges from conventional denoising in the diffusion framework. Inspired by the Masked Autoencoder (MAE) (He et al., 2022), we replace noise addition with a masking operation (see Fig. 1), introducing a new self-supervised pre-training paradigm for semantic segmentation named the masked diffusion model (MDM). Furthermore, we propose a simple yet impactful approach to bridge the gap between reconstructive pre-training tasks and downstream prediction tasks, i.e., substituting the commonly used Mean Squared Error (MSE) loss, prevalent in traditional denoising diffusion models and other pre-training models, with the Structural Similarity (SSIM) loss (Wang et al., 2004). Our approach achieved state-of-the-art accuracy, showing the potential of alternative corruption strategies and emphasizing the efficacy of the SSIM loss in diffusion-based pre-training.

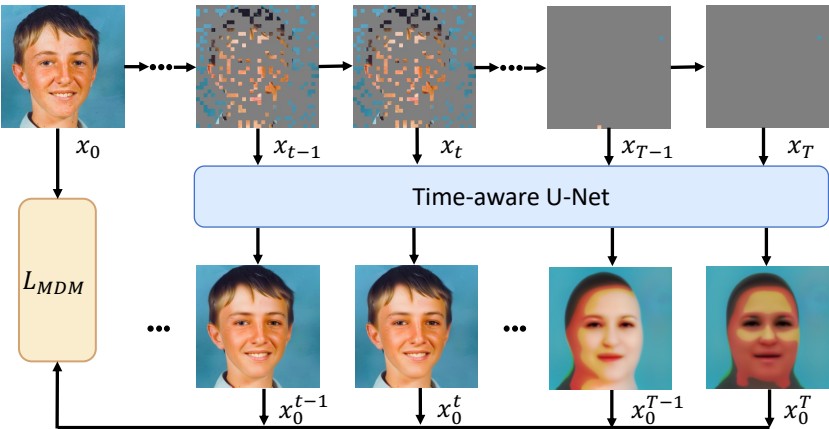

Figure 1: **Dynamic masking process.** Portions of data are probabilistically masked according to the timestep $t$ and subsequently reconstructed via a time-aware U-Net.

Our contribution can be summarized as follows: (1) We extend denoising diffusion models into a self-supervised pre-training algorithm, using denoising as a preliminary task for training a robust representation learning autoencoder. We empirically decompose the interrelation between the generative capability and representation learning ability inherent in diffusion models; the representation ability of diffusion models does not originate from their generative power. (2) We propose a novel paradigm called the masked diffusion model (MDM) for self-supervised representation learning (primarily tested for segmentation task), which fully abandons the theoretical guarantee of traditional generative diffusion models. In this model, we replace the conventional Gaussian noise in traditional diffusion models with a masking operation. We also utilize the Structural Similarity Index (SSIM) loss to minimize the disparity between the pre-training reconstruction task and downstream dense prediction task. (3) Extensive experiments in downstream semantic segmentation tasks show that MDM outperforms DDPM and other baselines on multiple datasets emcompassing both medical and natural images. Particularly noteworthy is the robust performance of MDM in few-shot scenarios.

## 2 RELATED WORK

### 2.1 DIFFUSION MODELS

Diffusion models (Sohl-Dickstein et al., 2015; Ho et al., 2020) are a class of generative models which have recently gained significant popularity. Diffusion models define a Markov chain of diffusion steps to gradually add noise to data and train a deep neural network to invert the diffusion process from a known Gaussian distribution. The more powerful architectures and advanced objectives proposed by Nichol & Dhariwal (2021); Dhariwal & Nichol (2021) further improve the generative quality and diversity of diffusion models.

Many recent works (Rissanen et al., 2023; Bansal et al., 2022; Hoogeboom & Salimans, 2023; Daras et al., 2022) attempted to propose alternative degradation mechanisms for diffusion models that aim to supplant the conventional additive Gaussian noise. Nevertheless, these works have either struggled to replicate the exceptional image quality achieved by traditional denoising diffusion (Bansal et al., 2022), or have resorted to blurring degradations, which essentially amounts to introducing Gaussian noise in the frequency domain (Rissanen et al., 2023; Hoogeboom & Salimans, 2023; Daras et al., 2022). Currently, there is no solid theoretical foundation or compelling empirical evidence to show that the Gaussian noise in diffusion models can be replaced. Our work revisits denoising diffusion models from a self-supervised representation learning perspective rather than from an image synthesis standpoint, and takes the first step toward removing Gaussian noise from diffusion.

### 2.2 SELF-SUPERVISED LEARNING

Self-supervised learning approaches aim to learn from unlabelled data via pre-text tasks (Doersch et al., 2015; Wang & Gupta, 2015; Pathak et al., 2017; Gidaris et al., 2018). Recently, the masked

autoencoder (MAE) (He et al., 2022) has shown remarkable capability as an effective pre-training strategy by reconstructing masked patches. Baranchuk et al. (2022) finds that denoising diffusion models (DDPM) learn semantic representations from a self-supervised denoising training process. (Lei et al., 2023) tries to combine MAE and DDPM to get a more powerful and faster image synthesizer, while our work focuses on the representation learning ability. Wei et al. (2023) proposes DiffMAE and evaluates it on downstream recognition tasks. DiffMAE still keeps denoising process in DDPM and conditions diffusion models on masked input. DiffMAE does not achieve better fine-tuning performance than MAE. In contrast, our work fully removes the additive Gaussian noise in diffusion and outperforms both MAE and DDPM in downstream semantic segmentation tasks.

## 3 BACKGROUND

### 3.1 DENOISING DIFFUSION PROBABILISTIC MODELS (DDPM)

Given a data distribution $x_0 \sim q(x_0)$, each step of the Markovian *forward process* (*diffusion process*) can be defined by adding Gaussian noise according to a variance schedule $\beta_1, ..., \beta_T$:

$$q(x_t|x_{t-1}) := \mathcal{N}(x_t; \sqrt{1-\beta_t}x_t, \beta_t I)$$

A key property of the forward process is that it allows sampling $x_t$ directly from a Gaussian distribution:

$$q(x_t|x_0) := \mathcal{N}(x_t; \sqrt{\bar{\alpha}_t}x_0, (1-\bar{\alpha}_t)I), \ x_t = \sqrt{\bar{\alpha}}x_0 + \epsilon\sqrt{1-\bar{\alpha}_t}, \epsilon \sim \mathcal{N}(0,I) \tag{1}$$

where $\alpha_t := 1 - \beta_t$ and $\bar{\alpha}_t := \prod_{s=1}^{s} \alpha_s$.

To sample from the data distribution $q(x_0)$, a reverse sampling procedure (the *reverse process*) can be employed by first sampling from $q(x_T)$ and then iteratively sampling reverse steps $q(x_{t-1}|x_t)$ until reaching $x_0$. By adopting the setting for $\beta_T$ and $T$ in Ho et al. (2020), the outcome is that $q(x_T) \approx \mathcal{N}(x_T; 0, I)$, which makes sampling $x_T$ trivial. The only thing left is to train a model $p_\theta(x_{t-1}|x_t)$ to approximate the unknown $q(x_{t-1}|x_t)$. Based on Bayes theorem, $q(x_{t-1}|x_t) = \frac{q(x_t|x_{t-1})q(x_{t-1})}{q(x_t)}$ is intractable since both $q(x_{t-1})$ and $q(x_t)$ are unknown. Song et al. (2021) shows that the true $q(x_s|x_t) \rightarrow q(x_s|x_t, \hat{x_0})$ as $s \rightarrow t$ for $s < t$. Therefore, the tractable posterior conditioned on $x_0$ can be considered as a compromise:

$$q(x_{t-1}|x_t, x_0) = \frac{q(x_t|x_{t-1})q(x_{t-1}|x_0)}{q(x_t|x_0)}$$
$$= \mathcal{N}(x_{t-1}; \widetilde{\mu}_t(x_t, x_0), \widetilde{\beta}_t I), \tag{2}$$

where $\widetilde{\mu}_t(x_t, x_0) := \frac{\sqrt{\bar{\alpha}_{t-1}}\beta_t}{1-\bar{\alpha}_t}x_0 + \frac{\sqrt{\alpha_t}(1-\bar{\alpha}_{t-1})}{1-\bar{\alpha}_t}x_t$, $\widetilde{\beta}_t := \frac{1-\bar{\alpha}_{t-1}}{1-\bar{\alpha}_t}\beta_t$. Assuming a well-trained neural network $f_\theta$ that estimates $x_0$ from $x_t$, the targeted $q(x_{t-1}|x_t)$ can be obtained by substituting the actual $x_0$ in Equation 2 with the estimated $\hat{x_0}$:

$$q(x_{t-1}|x_t) \approx q(x_{t-1}|x_t, \hat{x_0} = f_\theta(x_t, t))$$

Instead of directly predicting $x_0$, Ho et al. (2020) proposes an equivalent training strategy which trains a neural network $\epsilon_\theta(x_t, t)$ to predict the noise $\epsilon_t$ from Equation 1 to produce better samples. Once $\hat{\epsilon}_t = \epsilon_\theta(x_t, t)$ is available, $\hat{x_0}$ can be easily derived by:

$$\hat{x_0} = \frac{1}{\sqrt{\bar{\alpha}}}(x_t - \sqrt{1-\bar{\alpha}_t}\hat{\epsilon}_t)$$

The network is optimized by the commonly used simple objective:

$$\mathcal{L}_{\text{DDPM}} := E_{t\sim[1,T],x_0\sim q(x_0),\epsilon\sim\mathcal{N}(0,I)}[\|\epsilon - \epsilon_\theta(x_t, t)\|^2] \tag{3}$$

### 3.2 MASKED AUTOENCODERS (MAE)

MAE is a powerful self-supervised pre-training algorithm which learns representations by reconstructing the masked patches based on the visible patches. In detail, the full image is firstly divided into non-overlapping patches following Vision Transformers (ViT) (Dosovitskiy et al., 2021). Then a random subset of patches is sampled and used as input for the ViT-based MAE Encoder. The encoder embeds these visible patches by a linear projection and incorporates corresponding positional embeddings, then processes the resulting set using a sequence of Transformer blocks. The MAE decoder takes a full set of tokens consisting of patch-wise representations from the encoder

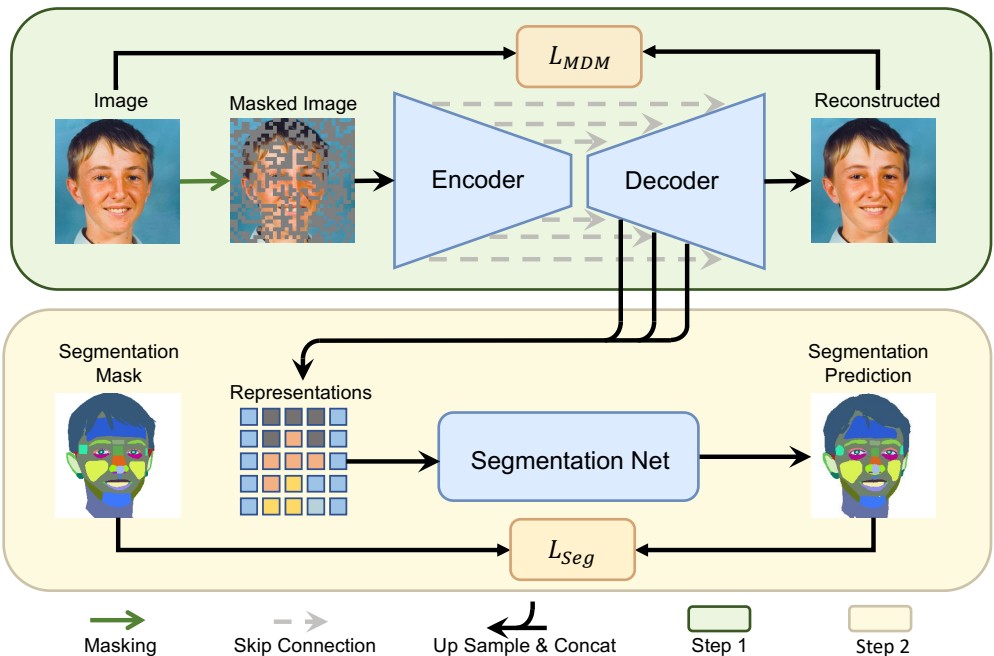

Figure 2: **Overview of our proposed method.** During pre-training, only the masked diffusion model (Encoder and Decoder) in Step 1 is trained. We partially mask the clean image based on a randomly sampled timestep $t$. The masked diffusion model then takes the masked image and reconstructs it. For the downstream segmentation task, the pre-trained model in Step 1 is frozen as a representation generator, and the segmentation network in Step 2 is trained with the representations extracted from Step 1.

and learnable mask tokens with positional embeddings. Since the decoder is only used during pre-training for the reconstruction task, it can be designed in a manner that is much lighter than the encoder. In MAE, the loss function is determined by calculating the mean squared error (MSE) between the previously masked portion of the reconstructed images and the corresponding original images at the pixel level. In particular, MAE employs the normalized pixels as the reconstruction target, leading to an improvement in representation quality.

## 4 METHODOLOGY

Our proposed masked diffusion model (MDM) is a variant of traditional denoising diffusion probabilistic models (DDPM) that reconstructs the whole image given partial observation. While DDPM serves as both an impressive image synthesizer and a robust self-supervised representation learner, MDM, in contrast, is solely designed for self-supervised representation learning. Following Baranchuk et al. (2022), the pixel-level representations extracted from a self-supervised pre-trained MDM are used for downstream dense prediction tasks, e.g., image segmentation. An overview of our method is shown in Figure 2. The following subsections will provide a detailed explanation of the pre-training process and how to apply the learned representations to the downstream tasks.

### 4.1 SELF-SUPERVISED PRE-TRAINING WITH MASKED DIFFUSION MODEL

According to Equation 3, DDPM is trained by constantly predicting the noises in the input noisy images. Well-trained DDPM produces high-quality representations for label efficient segmentation (Baranchuk et al., 2022). From the perspective of self-supervised pre-training, DDPM can be treated as a special kind of denoising autoencoder (DAE)[1]. DAE degrades an input signal under different corruptions and learns to reconstruct the original signal. In particular, DDPM corrupts the original

---

[1]In denoising autoencoders (DAE), the term "denoising" refers to restoring original data from data degraded by a certain corruption. Conversely, the term "denoising" in other places in this paper explicitly refers to the process of removing introduced Gaussian noise in noisy images and reconstructing the original images.

signal by adding a Gaussian noise controlled by a timestep $t$. Inspired by MAE and DDPM, our masked diffusion model (MDM) takes an image in which a random portion is masked based on the timestep $t$, and subsequently restores the image.

**Architecture.** Following guided diffusion (Dhariwal & Nichol, 2021), MDM uses the UNet architecture, which has been found to be highly effective for diffusion models. The UNet model consists of a series of residual layers and downsampling convolutions, followed by another set of residual layers with upsampling convolutions. The layers with the same spatial size are connected by skip connections. All the residual blocks follow the design in BigGAN (Brock et al., 2019). The model uses self-attention blocks at the $32 \times 32$, $16 \times 16$, and $8 \times 8$ resolutions. Diffusion step $t$ is embedded into each residual block.

**SSIM Loss.** Both MAE and DDPM calculate the Mean Squared Error (MSE) loss between the reconstructed and original images to assess the reconstruction quality. Nevertheless, we have observed that a generative model's ability to generate images with low MSE loss does not consistently guarantee that high-quality semantic representations can be extracted from the model. As a solution, we turn to Structural Similarity Index (SSIM) loss, which is also commonly employed in image restoration tasks. SSIM loss guides the reconstruction process to create images that are more similar to the original ones in terms of their structural information (Wang et al., 2004). By doing so, we aim to narrow the gap between the reconstruction task and the subsequent segmentation task, as structural information is often pivotal for accurate segmentation. The effectiveness of SSIM loss is further shown and discussed in ablation studies. MDM is finally optimized by the following objective:

$$\mathcal{L}_{\text{MDM}} := E_{t \sim [1,T], x_0 \sim q(x_0)}[\mathcal{L}_{\text{SSIM}}(x_0, \mathcal{U}_\theta(x_t, t))], \tag{4}$$

$$\mathcal{L}_{\text{SSIM}}(x, \hat{x}) := \frac{1 - \text{SSIM}(x, \hat{x})}{2},$$

$$\text{SSIM}(x, \hat{x}) := \frac{(2\mu_x \mu_y + c_1)(2\sigma_{xy} + c_2)}{(\mu_x^2 + \mu_y^2 + c_1)(\sigma_x^2 + \sigma_y^2 + c_2)},$$

where $c_1 = (k_1 L)^2$ and $c_2 = (k_2 L)^2$ are two variables to stabilize the division with weak denominator, with $L$ being the dynamic range of the pixel-values.

**Masking and Reconstructing.** MDM is trained by iterative masking and reconstructing until the model converges. We describe the detailed training procedure given a single image and one sampled timestep below. Given an image $x_0 \in \mathbb{R}^{H \times W \times C}$ sampled from the data distribution $q(x_0)$, we divide and reshape it into $N = \frac{H \times W}{P^2}$ patches denoted by $p_0^{(1)}, p_0^{(2)}, \ldots, p_0^{(N)}$. Each patch denoted by $p_0^{(i)}$ is represented as a vector with a size of $P^2 C$, where $P$ is the patch size and $C$ is the number of channels in the image. To initiate the diffusion process, we first sample a diffusion timestep $t$ from the interval $[1, T]$, while the masking ratio $R_m$ is defined as $R_m = \frac{t}{T+1}$. Then we randomly shuffle the list of patches and replace the last $\lfloor R_m \times N \rfloor$ patches with zero values. Afterward, we unshuffle the list to its original order, resulting in the corrupted image $x_t$. In accordance with the reverse procedure employed in DDPM, the UNet model denoted by $\mathcal{U}_\theta$ takes both the corrupted image $x_t$ and its corresponding timestep $t$ as input and subsequently generates an estimate for the intact image, denoted as $\hat{x}_0 = \mathcal{U}_\theta(x_t, t)$. The optimization of the $\mathcal{U}_\theta$ model is achieved using the $\mathcal{L}_{\text{MDM}}$ loss function defined in Equation 4.

## 4.2 DOWNSTREAM FEW-SHOT SEGMENTATION

In this paper, we explore a few-shot semi-supervised setting where we first pre-train MDM on a large unlabelled image dataset $\mathcal{X}_{\text{unlabelled}} = \{x_0^1, ..., x_0^N\} \in \mathbb{R}^{H \times W \times C}$ using the self-supervised approach as previously described. Then we leverage the features extracted from the MDM decoder to train a segmentation head $S_\phi$ (MLP) on a smaller dataset $\mathcal{X}_{\text{labelled}} = \{x_0^1, ..., x_0^M\} \in \mathbb{R}^{H \times W \times C}$ with K-class semantic labels $\mathcal{Y}_{\text{labelled}} = \{y_0^1, ..., y_0^M\} \in \mathbb{R}^{H \times W \times K}$. It is worth noting that $\mathcal{X}_{\text{labelled}}$ is not necessarily a subset of $\mathcal{X}_{\text{unlabelled}}$. Our experiments demonstrate that despite the distinction between $\mathcal{X}_{\text{unlabelled}}$ and $\mathcal{X}_{\text{labelled}}$, the features learned from $\mathcal{X}_{\text{unlabelled}}$ can still yield valuable benefits for accurately segmenting $\mathcal{X}_{\text{labelled}}$, as long as both datasets belong to the same domain.

To extract features for our analysis, we focus on a subset of UNet decoder blocks at a specific diffusion step $t$. We feed the labelled $\mathcal{X}_{\text{labelled}}$ and the diffusion timestep $t$ into the pre-trained

Diffusion $\mathcal{U}_\theta$, extracting the features based on the specified blocks setting $\mathcal{B}$. The extracted features are upsampled to match the image size and then concatenated, resulting in feature vectors $\mathcal{F}_{\text{labelled}} = \{f_{t,\mathcal{B}}^1, ..., f_{t,\mathcal{B}}^M\} \in \mathbb{R}^{H \times W \times C_f}$, where $C_f$ represents the number of channels, which varies depending on the blocks setting $\mathcal{B}$. The loss of the segmentation network $S_\phi$ is defined as follows:

$$\mathcal{L}_{\text{Seg}} := \frac{1}{M} \sum_i^M \text{CrossEntropy}(S_\phi(f_{t,\mathcal{B}}^i), y_0^i)$$

### 4.3 COMPARISONS WITH MAE AND DDPM

Our MDM differentiates itself from MAE and DDPM in three key aspects: (1) Purpose: While DDPM serves as a potent generative model and self-supervised pre-training algorithm, MDM and MAE concentrate on self-supervised pre-training. The generative ability of MDM can be further explored. (2) Architecture: MAE employs the Vision Transformer (ViT) architecture, whereas DDPM and our MDM utilize U-Net. (3) Corruption: DDPM introduces increasing additive Gaussian noise, MAE applies patch masking and only uses visible patches as input, and MDM employs a masking strategy guided by a sampled timestep $t$ from $[1, T]$, using the whole corrupted image as input.

## 5 EXPERIMENTS

We assess the effectiveness of our MDM on various datasets, spanning both medical images—the Gland segmentation dataset (GlaS) (Sirinukunwattana et al., 2017) and MoNuSeg (Kumar et al., 2017; 2020)—and natural images (FFHQ-34 and CelebA-19 (Baranchuk et al., 2022)), in the context of few-shot segmentation following the pre-training phase. FFHQ-34 is a subset of the FFHQ-256 dataset (Karras et al., 2019) with segmentation annotations, and we use FFHQ-256 to pre-train for FFHQ-34 and CelebA-19 segmentation. We adopt in-domain pre-training for all pre-training methods on GlaS and MoNuSeg. Comprehensive dataset details are provided in Appendix A.1. We also provide additional evaluation of MDM on the classification task in Appendix C.

### 5.1 COMPARISONS WITH STATE-OF-THE-ART METHODS

**Medical Image Datasets.** We compare our MDM with the two types of current state-of-the-art methods on GlaS and MoNuSeg, covering six traditional segmentation methods: UNet (Ronneberger et al., 2015), UNet++ (Zhou et al., 2018), Swin UNETR (Hatamizadeh et al., 2021), AttUNet (Wang et al., 2022b), UCTransNet (Wang et al., 2022a), MedT (Valanarasu et al., 2021) and two self-supervised pre-training methods, MAE and DDPM, that allow one to extract intermediate activations, much like in our method. For a fair comparison, all the methods not elaborated in Section A.2 adhere to their respective official configurations.

In Table 1, we report a comprehensive comparison of methods in terms of Dice, IoU, and AJI metrics on GlaS and MoNuSeg across two distinct scenarios. The qualitative examples of segmentation with our method and other comparable models are shown in Fig. 3 and Fig. 4. Notably, in Fig. 4, we have marked regions of superior performance by our MDM with red boxes. The segmentation network, trained utilizing our proposed MDM representations, yields significant performance improvements over previous techniques for both GlaS and MoNuSeg. Furthermore, all the traditional segmentation methods fail to provide reasonable segmentation predictions when operating with only 10% of the available labels for training. In contrast, our MDM not only surpasses the accuracy achieved by state-of-the-art self-supervised methods such as MAE and DDPM, but also attains accuracy levels close to those achieved with a full label set while using only 10% of the labels.

**Natural Image Datasets.** We compare MDM with two categories of state-of-the-art methods on FFHQ-34 and CelebA-19, including two methods that employ extensive annotated synthetic image sets for segmentation training: DatasetGAN (Zhang et al., 2021), DatasetDDPM (Baranchuk et al., 2022) and four self-supervised methods similar to our approach: SwAV, SwAVw2 (Caron et al., 2020), MAE and DDPM.

Table 2 presents the experimental results in terms of mIoU on FFHQ-34 and CelebA-19. Our MDM outperforms the previous state-of-the-art results. Additionally, we visualize the segmentation results in Figure 5. In particular, MDM is the only method which correctly classifies the hat in the third row image, which proves the representations extracted from MDM are semantically rich.

Table 1: **Comparisons with previous methods on GlaS and MoNuSeg.** The results are presented in the format of "mean±std". Both DDPM and MDM are trained for 10000 iterations on GlaS and 5000 iterations on MoNuSeg. We present the results for two different scenarios: one with 100% segmentation labels and the other with only 10% segmentation labels.

| Method | GlaS 100% (85) | | GlaS 10% (8) | | MoNuSeg 100% (30) | | MoNuSeg 10% (3) | |
|---|---|---|---|---|---|---|---|---|
| | Dice (%) | IoU (%) | Dice (%) | IoU (%) | Dice (%) | AJI (%) | Dice (%) | AJI (%) |
| UNet | 85.93±2.92 | 75.44±4.42 | 53.46±11.09 | 37.24±10.06 | 74.56±0.98 | 60.22±1.31 | 54.79±2.50 | 41.06±2.14 |
| UNet++ | 86.62±0.99 | 76.41±1.56 | 77.05±2.11 | 62.72±2.7 | 80.33±0.69 | 67.30±0.94 | 74.26±1.62 | 59.49±1.98 |
| Swin UNETR | 86.74±1.32 | 76.61±2.05 | 73.95±2.00 | 58.71±2.54 | 79.99±0.42 | 66.81±0.57 | 72.47±2.72 | 57.36±3.27 |
| AttUNet | 86.12±1.99 | 75.68±3.04 | 66.35±4.79 | 49.83±5.16 | 79.74±1.03 | 66.55±1.36 | 70.87±3.46 | 55.72±3.84 |
| UCTransNet | 85.10±2.44 | 74.14±3.65 | 55.87±5.61 | 38.98±5.44 | 78.80±1.15 | 65.57±1.34 | 64.33±5.19 | 48.78±5.35 |
| MedT | 81.02±2.10 | 68.14±2.98 | 59.46±11.31 | 43.18±10.86 | 77.55±1.02 | 63.48±1.35 | 64.49±2.87 | 50.60±3.09 |
| MAE | 89.71±0.92 | 81.35±1.51 | 88.56±0.67 | 79.47±1.08 | 73.68±1.48 | 58.62±1.78 | 76.19±0.18 | 61.61±0.23 |
| DDPM | 90.45±0.37 | 82.56±0.61 | 90.30±0.47 | 82.32±0.77 | 80.31±0.58 | 67.31±0.77 | 74.37±3.08 | 60.03±3.48 |
| **MDM(ours)** | **91.95±1.25** | **85.13±2.09** | **91.60±0.69** | **84.51±1.15** | **81.01±0.35** | **68.25±0.49** | **79.71±0.75** | **66.43±1.02** |

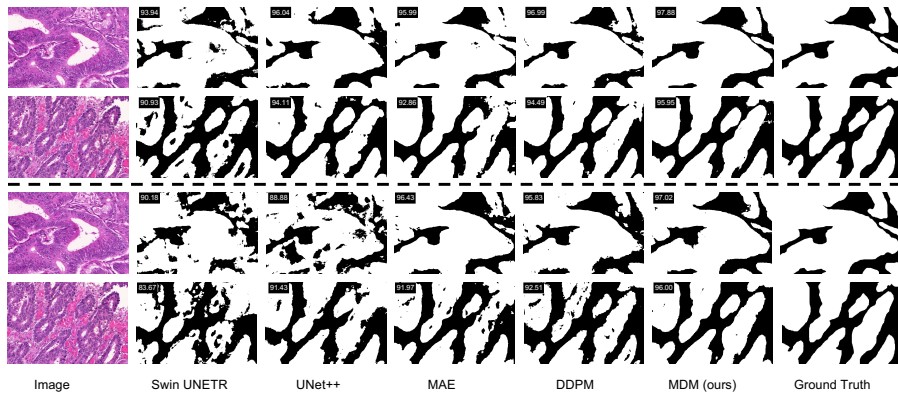

Figure 3: **Qualitative Visualization** on GlaS test sets under full training labels setting (first 2 rows) and 10% training labels setting (last 2 rows) with the dice score (%) of each prediction.

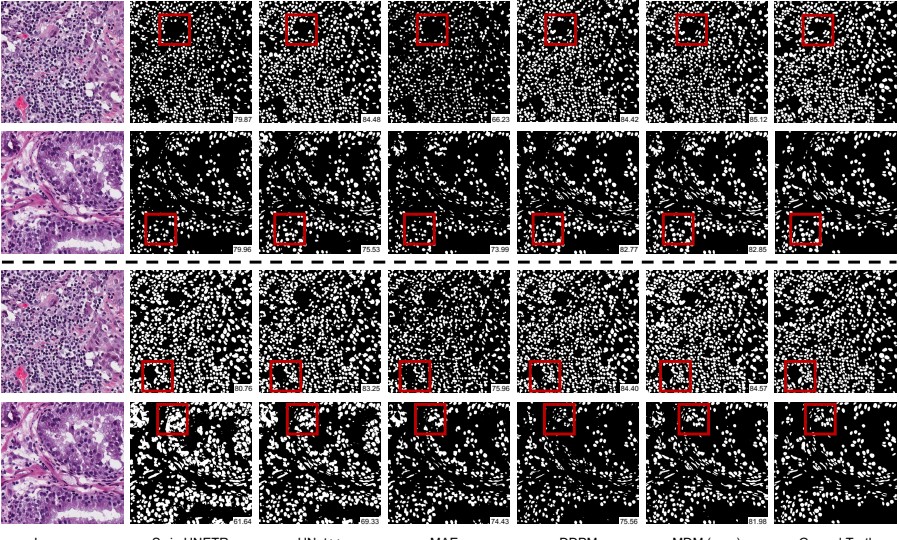

Figure 4: **Qualitative Visualization** on MoNuSeg test sets under full training labels setting (first 2 rows) and 10% training labels setting (last 2 rows) with the dice score (%) of each prediction.

**Key Observations.** We summarize the key observations as follows: (1) The proposed MDM outperforms DDPM across all four datasets under different settings, which indicates that the denoising in diffusion is not irreplaceable, at least from a self-supervised representation learning perspective. (2) MDM achieves better performance than MAE, even though both methods employ similar mask-

Table 2: **Semantic segmentation performance on FFHQ-34 and CelebA-19 in terms of mIoU.** Symbol $^*$ denotes the evaluation of CelebA-19 utilizing models pre-trained on FFHQ-256. MDM is pre-trained for 40000 iterations for FFHQ-34 and CelebA-19, while other pre-training methods use the checkpoints provided by Baranchuk et al. (2022).

| Method | FFHQ-34 | CelebA-19$^*$ |
|---|---|---|
| DatasetGAN | 55.52±0.24 | - |
| DatasetDDPM | 55.01±0.27 | - |
| SwAV | 52.53±0.13 | 51.99±0.20 |
| SwAVw2 | 54.57±0.16 | 51.17±0.14 |
| MAE | 57.06±0.20 | 57.27±0.08 |
| DDPM | 59.36±0.17 | 58.86±0.12 |
| **MDM(ours)** | **60.34±0.15** | **59.57±0.13** |

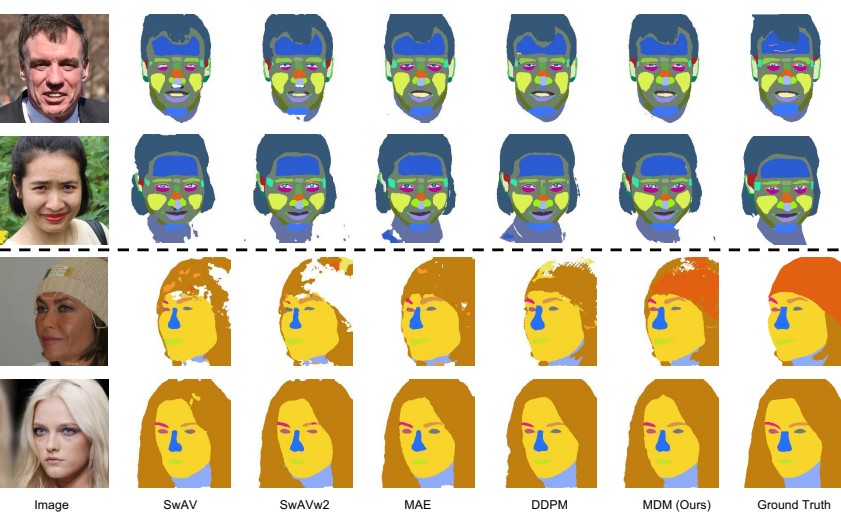

| Image | SwAV | SwAVw2 | MAE | DDPM | MDM (Ours) | Ground Truth |

Figure 5: **Qualitative Visualization** on FFHQ-34 (first 2 rows) and CelebA-19 (last 2 rows).

ing and restoration pre-training techniques. We attribute this improvement to MDM's utilization of different levels of masking ($T$ levels). This contributes to the acquisition of more robust and semantically meaningful representations, enhancing segmentation results. (3) MDM excels in few-shot scenarios, while conventional segmentation methods struggle. This positions MDM as a competitive choice for tasks with limited labeled data. Notably, this advantage is particularly relevant in medical datasets, where acquiring pixel-level labels is costly. This makes MDM's label-efficient characteristic beneficial for such data.

## 5.2 ABLATION STUDIES

Our ablation studies are based on 20000 iterations pre-training of DDPM and MDM for FFHQ-34 and 10000 iterations pre-training for GlaS. If not specified, other settings follow the implementation details in Appendix A.2. Here, we provide our ablation studies focusing on diffusion, loss functions, and reconstruction targets. Comprehensive ablation studies can be found in Appendix B.

**Diffusion.** Table 3 studies the impact of diffusion on our MDM and DDPM pre-training. Diffusion models consist of $T$ timesteps, each corresponding to an incremental level of corruption. We fix $t$ to 250 for DDPM pre-training and 50 for MDM pre-training. Then DDPM degrades to a vanilla denoising autoencoder and MDM degrades to a vanilla masked autoencoder (with a slight difference from MAE). The degraded DDPM and MDM show similar performance compared with MAE, while with diffusion, both DDPM and MDM outperform MAE. Surprisingly, MDM demonstrates a more substantial improvement over DDPM with diffusion and achieves the highest accuracy, even though MDM is not a standard diffusion model with powerful generation capability. This phenomenon further supports our conjecture: the efficacy of semantic representations derived from DDPM is not

Table 3: **Ablation results of diffusion.** Fixed $t$ means a diffusion model degrades to a vanilla autoencoder with a fixed level of corruption.

| Method | Fixed $t$ | GlaS 100% (85) | | GlaS 10% (8) | |
|--------|-----------|----------------|----------------|----------------|----------------|
| | | Dice (%) | IoU (%) | Dice (%) | IoU (%) |
| MAE | - | 89.71±0.92 | 81.35±1.51 | 88.56±0.67 | 79.47±1.08 |
| DDPM | ✓ | 89.82±0.41 | 81.52±0.76 | 87.10±0.67 | 77.15±1.06 |
| | × | **90.45±0.37** | **82.56±0.61** | **90.30±0.47** | **82.32±0.77** |
| MDM | ✓ | 88.68±0.54 | 79.67±0.86 | 86.82±1.04 | 76.72±1.59 |
| | × | **91.95±1.25** | **85.13±2.09** | **91.60±0.69** | **84.51±1.15** |

Table 4: **Ablation results of loss functions and reconstruction targets in terms of mIoU.** [†]: DDPM predicts the original image $x_0$ instead of the noise $\epsilon_t$.

| Method | Loss Type | GlaS 10% (8) | FFHQ-34 |
|--------|-----------|--------------|---------|
| DDPM | MSE | 82.32±0.77 | 58.75±0.16 |
| DDPM[†] | MSE | 78.77±1.01 | 51.63±0.16 |
| | **SSIM** | **81.79±1.13** | **56.97±0.18** |
| MAE | **MSE** | **79.47±1.08** | **57.06±0.20** |
| | SSIM | 76.72±1.75 | 56.55±0.13 |
| MDM | MSE | 82.70±0.79 | 55.06±0.21 |
| | **SSIM** | **84.51±1.15** | **59.18±0.11** |

solely contingent on its generative prowess. Instead, we can view DDPM as a denoising autoencoder with $T$ levels of noises, and it can potentially be substituted with other forms of random corruptions.

**SSIM Loss and Reconstruction Targets.** The effectiveness of SSIM loss and reconstruction targets is studied in Table 4. When applied to MDM and DDPM[†] pre-training, SSIM loss improves the accuracy on both GlaS and FFHQ-34 datasets compared to MSE. Meanwhile, SSIM loss does not work well in MAE. These findings highlight that choosing a suitable loss function is a straightforward yet effective way for minimizing the discrepancy between pre-training-acquired representations and those essential for subsequent segmentation tasks. Additionally, we investigate different reconstruction targets for DDPM pre-training. DDPM pre-trained with the noise prediction strategy achieves higher accuracy in downstream segmentation tasks compared to using the image prediction strategy. Our findings are consistent with previous research (Ho et al., 2020), indicating that predicting the noise $\epsilon_t$ can lead to improved quality in diffusion models, as opposed to predicting the clean image $x_0$. Thus, all DDPM models in this study are trained by predicting noise.

## 6 DISSCUSION

**Summary.** In this study, we revisit the denoising diffusion probability models (DDPM) within the framework of self-supervised pre-training and break down the relation of semantic representation quality and generation capability of DDPM. We introduce a novel pre-training paradigm for semantic segmentation by stripping noise from diffusion models and integrating masking operations akin to MAE. Furthermore, we demonstrate the effectiveness of SSIM loss in narrowing the gap between the reconstruction pre-training task and the downstream dense prediction task. Our proposed masked diffusion model (MDM) with SSIM loss achieves state-of-the-art performance in semantic segmentation on multiple benchmark datasets. The label-efficient attribute of MDM holds promising prospects for diverse few-shot dense prediction tasks.

**Limitations and Future Work.** (1) Our MDM implementation is currently applied exclusively to U-Net, with an evaluation focused on downstream semantic segmentation performance across two medical datasets and two natural image datasets. Future exploration could involve extending pre-training to more complex networks (e.g., ConvNeXt) using MDM, and assessing performance on popular, large-scale segmentation datasets. (2) The choice of SSIM loss over MSE loss in MDM aims to enhance semantic representations for segmentation; however, alternative optimization objectives tailored to specific datasets and downstream tasks can be explored in future research. (3) In this paper, we focus on the image domain. Exploring the adaptation of MDM to other data domains (e.g., audio) can be an interesting direction of further investigation.

## REPRODUCIBILITY STATEMENT

In the main text, we provide the necessary details for the implementation of our MDM. We describe the four datasets for evaluation, and the detailed experimental setting in appendix. We will release our code and pre-trained model checkpoints upon acceptance.

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

# A    Experimental Details

## A.1    Datasets

We show a comprehensive overview of datasets used in this paper in Table 5.

Table 5: **A comprehensive overview of each dataset.** "Training" and "Test" columns denote available image-label pairs for the downstream segmentation task, while "Classes" specifies the count of semantic classes. The "Pre-training" column indicates the number of images utilized during the self-supervised pre-training phase. Notably, for the CelebA-19 segmentation, the pre-trained models for FFHQ-34 are directly used. This specific configuration is intentionally chosen to rigorously assess the generalization capabilities of models.

| Dataset | Training | Test | Classes | Pre-training |
|---------|----------|------|---------|--------------|
| GlaS | 85 | 80 | 2 | 165 |
| MoNuSeg | 30 | 14 | 2 | 44 |
| FFHQ-34 | 20 | 20 | 34 | 70000 |
| CelebA-19 | 20 | 500 | 19 | - |

## A.2    Implementation Details

We leverage the MONAI library[2] (Cardoso et al., 2022) implementations for UNet, UNet++, Swin UNETR, and AttUNet, while the remaining baseline models are evaluated using their official implementations[3][4]. The training was performed on one Tesla V100 GPU with 16 GB memory. We train MAE, DDPM and MDM ourselves on GlaS and Monuseg. For FFHQ-256, if not specified, we only train MDM ourselves and use the pre-trained MAE, DDPM, SwAV and SwAVw2 in Baranchuk et al. (2022)[5].

In pre-training, we use the 165 training and test images for GlaS and the 44 training and test images for MoNuSeg. We use 70000 unlabelled $256 \times 256$ images from FFHQ-256 dataset for FFHQ-34 and CelebA-19. We randomly crop $256 \times 256$ patches as input for GlaS and MoNuSeg to make sure all the models are trained with $256 \times 256$ images. We only take random flip in pre-training. The patch size in MDM and MAE is set to 8. The batch size is 128 and the maximum diffusion step $T$ is 1000 for DDPM and MDM. All the other settings for DDPM and MDM follow the official implementation of guided diffusion[6] (Dhariwal & Nichol, 2021), with MAE following the official MAE implementation[7] (He et al., 2022).

Then, we evaluate the pre-trained models across the four datasets for few-shot segmentation. In particular, for CelebA-19, we directly use the MDM (along with other pre-trained models) trained with FFHQ-256 to evaluate our method's generalization ability. For DDPM and MDM evaluations, we use the pixel-level representations from the middle blocks $\mathcal{B} = \{8, 9, 10, 11, 12\}$ and $\mathcal{B} = \{5, 6, 7, 8, 9, 10, 11, 12\}$ of the 18 decoder blocks for two medical image datasets and two natural image datasets respectively. We set the diffusion step to $t = 250$ for DDPM on medical datasets and $t = 50$ for MDM on all four datasets, while the settings outlined in Baranchuk et al. (2022) guided other methods and datasets. The selection strategy will be elaborated upon in Appendix B.

The segmentation network (MLP) undergoes training until the point of convergence, defined by the cessation of decreasing training loss over a specific number of steps, utilizing the Adam optimizer with a learning rate of 0.001. While dealing with medical datasets, a batch size of 65536 is employed, whereas for FFHQ-34 and CelebA-19, a batch size of 64 is adopted in accordance with Baranchuk et al. (2022). Each batch is constituted of the representation of one pixel. Notably, no data augmentation is applied during the training of the segmentation network.

---

[2]https://github.com/Project-MONAI/MONAI

[3]https://github.com/McGregorWwww/UCTransNet

[4]https://github.com/jeya-maria-jose/Medical-Transformer

[5]https://github.com/yandex-research/ddpm-segmentation

[6]https://github.com/openai/guided-diffusion

[7]https://github.com/facebookresearch/mae

Table 6: **Ablation results of different timesteps in terms of Dice and IoU.** 50+150+250 represents concatenating representations extracted at $t = 50, 150, 250$.

| Method | Timestep | GlaS 100% (85) | | GlaS 10% (8) | | Average | |
|---|---|---|---|---|---|---|---|
| | | Dice (%) | IoU (%) | Dice (%) | IoU (%) | Dice (%) | IoU (%) |
| DDPM | 0 | 88.96±0.54 | 80.13±0.87 | 87.98±0.51 | 78.54±0.81 | 88.47±0.72 | 79.34±1.16 |
| | 50 | 90.40±0.24 | 82.47±0.39 | 88.54±0.77 | 79.43±1.23 | 89.47±1.10 | 80.95±1.79 |
| | 150 | 90.54±0.40 | 82.71±0.67 | 89.52±0.94 | 81.35±1.52 | 90.03±0.88 | 0.87±1.34 |
| | **250** | 90.45±0.37 | 82.56±0.61 | **90.30±0.47** | **82.32±0.77** | **90.38±0.42** | **82.44±0.69** |
| | 650 | 90.24±0.41 | 82.24±0.72 | 89.98±0.54 | 81.85±1.15 | 90.11±0.49 | 82.05±0.96 |
| | 50+150+250 | **90.76±0.44** | **83.10±0.75** | 89.06±0.53 | 80.21±1.18 | 89.91±0.99 | 81.66±1.77 |
| MDM | 0 | 91.93±1.50 | 85.11±2.51 | 91.32±0.84 | 84.05±1.42 | 91.63±1.22 | 84.58±2.06 |
| | **50** | 91.95±1.25 | 85.13±2.09 | **91.60±0.69** | **84.51±1.15** | **91.78±1.00** | **84.82±1.67** |
| | 150 | 91.98±1.21 | 85.17±2.05 | 91.49±0.73 | 84.32±1.23 | 91.74±1.00 | 84.75±1.70 |
| | 250 | 91.55±1.63 | 84.46±2.70 | 90.97±0.60 | 83.43±1.01 | 91.26±1.23 | 83.95±2.05 |
| | 650 | 91.87±0.93 | 84.98±1.57 | 90.45±1.13 | 82.58±1.88 | 91.16±1.24 | 83.78±2.09 |
| | 50+150+250 | **92.02±0.56** | **85.23±0.96** | 91.23±0.87 | 83.88±1.46 | 91.63±0.82 | 84.56±1.39 |

During inference, we use sliding window with size $256 \times 256$ to construct the segmentation predictions, ensuring alignment with the original image dimensions for both medical datasets. To make the results on the small datasets more convincing, we undertake 10 runs for all experiments, wherein the random seed is systematically varied from 0 to 9. We report the mean and std of the results. For GlaS we use dice coefficient (Dice) and Intersection over Union (IoU) as the evaluation metrics following UCTransNet (Wang et al., 2022a) while for MoNuSeg we report the Dice and Aggregated Jaccard Index (AJI) following the official challenge[8] (Kumar et al., 2017; 2020). For FFHQ-34 and CelebA-19, we quantify performance using the mean Intersection over Union (mIoU) following Baranchuk et al. (2022).

## B    ADDITIONAL ABLATION STUDIES

**Diffusion Timesteps.** We investigate the segmentation performance of representations extracted from MDM and DDPM across different diffusion timesteps $t$, as presented in Table 6. On average, the best results of DDPM are observed at $t = 250$ for the GlaS dataset, while MDM shows its highest performance with $t$ set to 50. The concatenation of three timesteps (50, 150, 250) does not always yield substantial enhancements but necessitates a $3\times$ increase in computational resources.

In light of these findings, we select $t = 250$ for DDPM on GlaS and MoNuSeg. Furthermore, we adhere to the official configuration in Baranchuk et al. (2022) wherein a concatenation of three timesteps (50, 150, 250) is employed for the FFHQ-34 and CelebA-19 datasets. For MDM, we set $t$ to 50 across all four datasets. We intentionally did not perform tuning for $t$ on every dataset.

**Blocks.** We display the results of k-means clustering ($k = 5$) for the frozen features of both DDPM and MDM in Figure 6. The features from the deeper layers (blocks with smaller values) exhibit coarse semantic masks, while those from shallower layers (blocks with larger values) reveal finer details yet lack the same level of semantic coherence for coarse segmentation. Therefore, we choose the middle blocks $\mathcal{B} = \{8, 9, 10, 11, 12\}$ and $\mathcal{B} = \{5, 6, 7, 8, 9, 10, 11, 12\}$ among the 18 decoder blocks for two medical image datasets and two natural image datasets respectively. This block configuration, as adopted in this study, has led to a slight enhancement in segmentation accuracy when using DDPM on FFHQ-34, in contrast to the official settings mentioned in Baranchuk et al. (2022). Importantly, we deliberately avoided tuning the blocks for each dataset individually.

A particularly intriguing observation is that due to the distinct reconstruction objectives of DDPM and MDM (noise for DDPM and image for MDM), the shallowest blocks (e.g., Block 16 and Block 17) of MDM yield meaningful semantic representations, whereas the same blocks for DDPM produce noise. For a fair comparison with DDPM, we did not use these shallow blocks for MDM. However, in practical scenarios, the judicious selection of blocks holds potential to further enhance the overall performance of our model.

---

[8]https://monuseg.grand-challenge.org

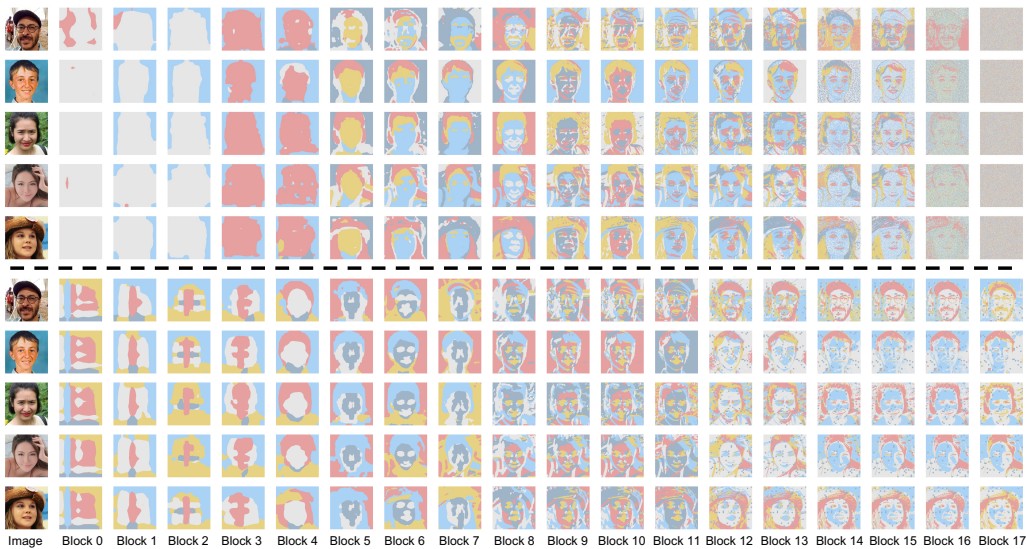

Figure 6: **K-means clustering of features** extracted from the UNet decoders of DDPM (first 5 rows) and MDM (last 5 rows) on FFHQ-34 at $t = 50$.

Table 7: **Ablation results showcasing the impact of different patch sizes on MDM.**

| Method | Patch Size | GlaS 100% (85) | | GlaS 10% (8) | |
|---|---|---|---|---|---|
| | | Dice (%) | IoU (%) | Dice (%) | IoU (%) |
| | 4 | 90.61±0.42 | 82.83±0.71 | 90.41±0.76 | 82.51±1.28 |
| MDM | **8** | **91.95±1.25** | **85.13±2.09** | **91.60±0.69** | **84.51±1.15** |
| | 16 | 90.97±0.60 | 83.44±1.01 | 90.60±0.41 | 82.82±0.69 |

**Patch Sizes.** We compare different patch sizes for MDM in Table 7. MDM achieves the best segmentation results when patch size is 8 (our final value). Note that even the performance of MDM experiences a marginal decrease for patch sizes of 4 or 16, it still surpasses the performance of existing methods.

**Training Schedules.** The impact of the training schedule length on downstream segmentation for MDM is illustrated in Figure 7. The accuracy exhibits a steady increase as the training duration extends, which suggests the scalability of MDM when applied to substantial unlabeled datasets.

**Robustness.** We investigate the robustness of our model against corrupted data in Figure 8. We first train our model on clean data. Then we apply 15 types of corruption from Hendrycks & Dietterich (2019) to the test images (Gaussian Noise, Shot Noise, Impulse Noise, Defocus Blur, Frosted Glass Blur, Frost, Brightness, Contrast, Elastic, Pixelate, JPEG, Speckle, Gaussian Blur, Spatter and Saturate). We report the average IoU for each method tested under different levels of corruption. MDM demonstrates strong robustness against various types of corruption, even without employing any data augmentation related to the applied corruption (e.g., noise) during both pre-training and downstream segmentation training.

## C ADDITIONAL CLASSIFICATION TASK

We evaluate our MDM with other self-supervised methods using linear probing on CIFAR-10 (Krizhevsky, 2009), where 50000 images are used for training and 10000 images are used for testing. MDM is pre-trained by a self-supervised way, and the representations are extracted as the input of one fully connected layer for classification. We retrieve the results of MoCov1 and MoCov2 (He et al., 2019; Chen et al., 2020c) from Radford et al. (2021) and the results of SimCLR, SimCLRv2 (Chen et al., 2020a;b) and DDPM from Xiang et al. (2023). Our MDM achieves much better linear

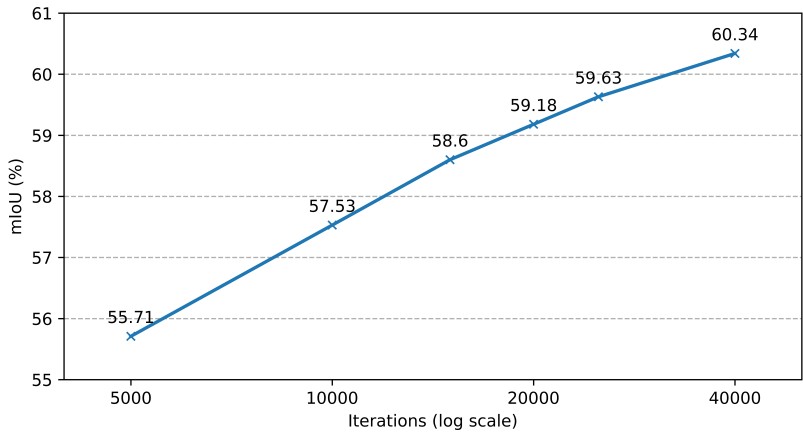

Figure 7: **MDM training schedules.** FFHQ-256 is used for MDM pre-training and FFHQ-34 is used for downstream segmentation.

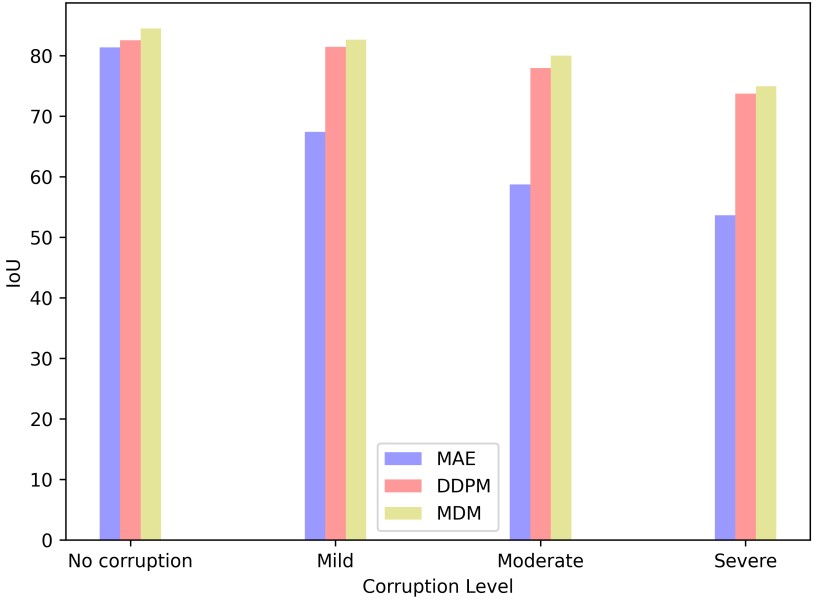

Figure 8: **Robustness Analysis** on GlaS test set in terms of IoU.

evaluation accuracy than DDPM which has the same learning type and network and competitive performance compared to the contrastive learning methods. This suggests that our MDM can also generalize to other downstream tasks, not limited to the semantic segmentation task alone.

Table 8: **Comparisons with self-supervised methods on CIFAR10.**

| Learning Type | Method | Network | Acc (%) |
|---|---|---|---|
| Contrastive | MoCov1 | ResNet-50 | 85.0 |
| | MoCov2 | ResNet-50 | 93.4 |
| | SimCLR | ResNet-50 | 94.0 |
| | SimCLRv2 | ResNet-101 | 96.4 |
| Generative | DDPM | U-Net | 90.1 |
| | MDM | U-Net | 94.8 |

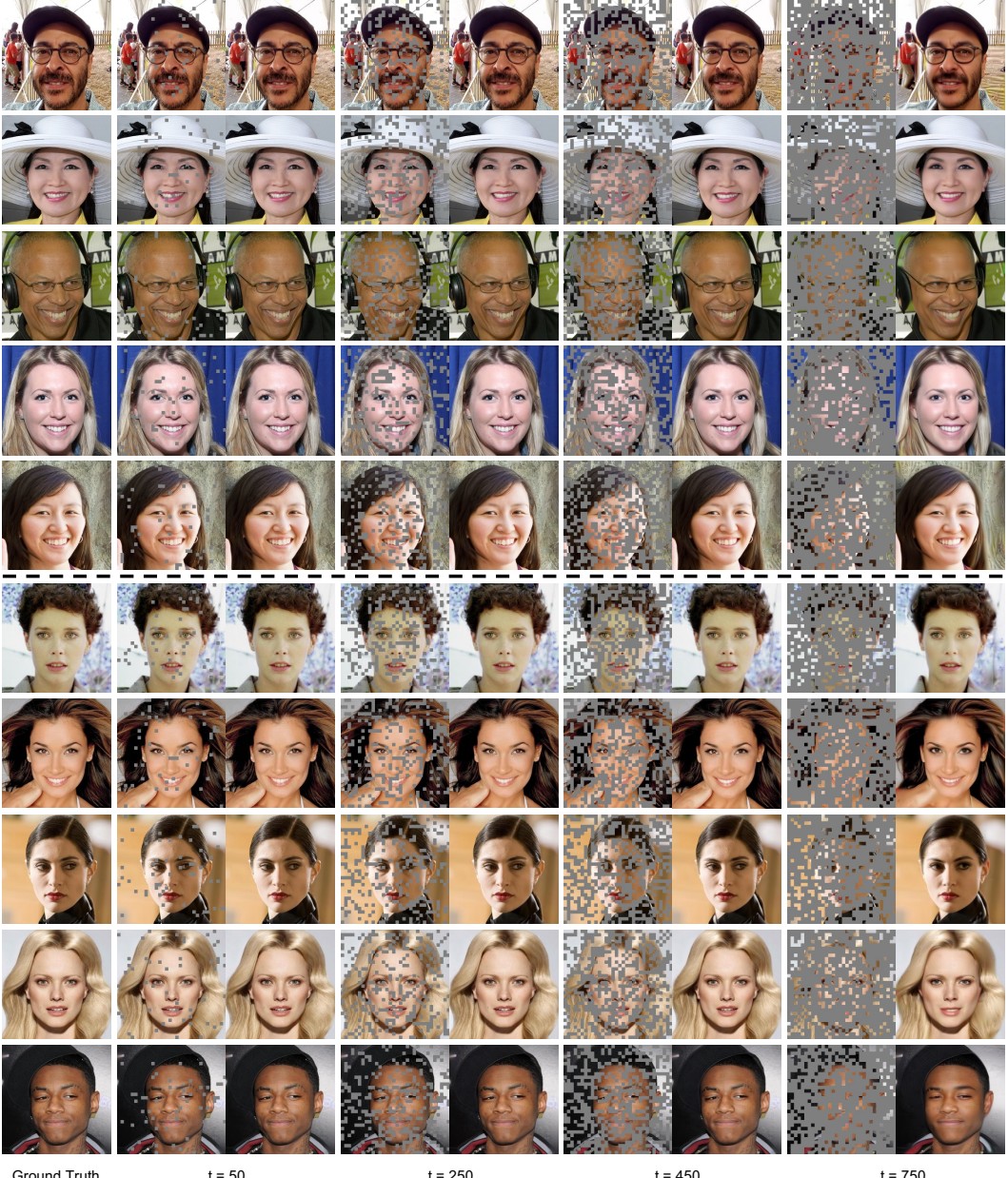

Figure 9: **Visualizations of MDM reconstruction** on FFHQ-34 test set (first 5 rows) and CelebA-19 test set (last 5 rows), using MDM trained on FFHQ-256 for 40000 iterations. The results at $t =$ 50, 250, 450, and 750 are presented. A greater value of $t$ corresponds to a higher masking ratio. For each pair, we show the masked image (left) and our MDM reconstruction (right).

## D ADDITIONAL QUALITATIVE RESULTS

The results of reconstructed images using the MDM technique on the FFHQ-34 and CelebA-19 datasets are shown in Figure 9. Importantly, MDM achieves successful reconstruction even on previously unseen images from the CelebA-19 dataset, using a substantial masking ratio of 75%.

