# OpenReview forum: "Masked Diffusion as Self-supervised Representation Learner"
_ICLR.cc/2024/Conference — Submitted to ICLR 2024_

### Official Review · Reviewer_caGX · 2023-10-31

**Soundness:** 4 excellent
**Presentation:** 4 excellent
**Contribution:** 3 good
**Rating:** 6
**Confidence:** 4

**Summary:**

The paper proposes masked diffusion model (MDM) for self-supervised learning. Masked diffusion is inspired by the use of the denoising diffusion as a representation learning approach. However, instead of using different levels of Gaussian noise, it uses masks, which at different timesteps of the forward process, blocks out more of the image. The inverted process is then learning to undo this masking operation.
The ability of masked diffusion to learn a useful representation for image segmentation is investigated on public datasets of both medical images (GlaS and MoNuSeg) as well as natural images (FFHQ-34 and CelebA-19) and compared both to standard supervised approaches (recent variations of convolutional and transformer based models) and to representation learning approaches based on the Masked AutoEncoder (MAE), the standard denoising diffusion probabilistic model (DDPM) and self-supervision approaches such as SwAV.

**Strengths:**

- Self-supervised learning is highly relevant and the manuscript is well written and easy to read.

- The approach appears to be novel and while the idea is perhaps not extremely original given prior work, the contribution is still solid.

- The approach is compared to a large number of other relevant approaches and consistently appears to do best, whether in the fully supervised case where 100% of the data is used for training the downstream segmentation task or in a more limited case where only 10% of the data is available for training the downstream segmentation task.

- The experiments appear to be easy to reproduce as the authors plan to release the code on acceptance.

**Weaknesses:**

- "This paper decomposes the interrelation between the generative capability and representation learning ability inherent in diffusion models." - I am not really sure what is meant by this or why it is significant.

- "a scalable self-supervised representation learner..." - What is meant or referred to with scalable here? It does not appear to be mentioned elsewhere.

- "While the prediction task demands a focus on high-level, low-frequency structural aspects of images," - I would have liked this to be better explored or more well supported to make statements such as this.

- "the representation ability of diffusion models does not originate from their generative power." - I don't see where this is investigsted in the proposed manuscript. E.g. the generative power of the proposed model is not shown.

- Will trained model weights also be released on acceptance?

- I am concerned by the apparent lack of a validation set. There seems to be a number of hyper parameters involved such as the choice of timesteps, blocks, patch sizes and training schedules covered in section B. Other choices such as loss function (SSIM or MSE) were also made. It is unclear if test sets were involved in these choices and if so can we expect the reported results to generalize?

**Questions:**

- See the question in the above related to the use of validation sets.

---

> ### Author Response · Authors · 2023-11-20
> **Response to Reviewer caGX**
>
> We would like to thank you for your positive evaluation of our work and providing valuable and insightful comments. Please find our responses to your comments in what follows:
>
> **Q1:**
> "This paper decomposes the interrelation between the generative capability and representation learning ability inherent in diffusion models." - I am not really sure what is meant by this or why it is significant.
>
> We apologize for the confusion.
>
> Recent works [1, 2, 3] which use diffusion models as representation learners usually assume that the superior generative quality of diffusion models supports them as meaningful discriminative learning methods. However, in this paper, we analyze the diffusion models from the self-supervised learning perspective and extend denoising diffusion models into a self-supervised pre-training algorithm, using denoising as a preliminary task. Then, we do not need to stick to denoising, as we did in our paper, if our objective is representation learning rather than image synthesis. This conclusion is significant since it allows researchers not to focus too much on the generative performance of diffusion models if their goal is to acquire meaningful representations. Instead, future works can explore various corruptions and adapt them to the diffusion framework for various data and downstream tasks. In practice, many works [1, 4, 5] which use extracted diffusion representations for downstream segmentation tasks can be potentially improved.
>
> **Q2:**
> "a scalable self-supervised representation learner..." - What is meant or referred to with scalable here? It does not appear to be mentioned elsewhere.
>
> We are sorry not making this information clear enough. The scalibility of MDM has been discussed in Appendix B of the original paper. We find that the representations learned by MDM continue to improve with longer training (Figure 7), which suggests that MDM is a scalable approach.
>
> **Q3:**
> While the prediction task demands a focus on high-level, low-frequency structural aspects of images," - I would have liked this to be better explored or more well supported to make statements such as this.
>
> Thanks for pointing this out. We revised this statement in the manuscript to make it clearer. We made the statement in the context of the few-shot scenarios where a lightweight network is used to avoid over-fitting. Due to the limited model capacity and labels, the model is not able to learn a good dense prediction if it has to process low-level, high-frequency details of images (e.g., texture). Therefore, it is essential that MDM learns high-level, low-frequency structural information in pre-training and provides the learned meaningful representations during fine-tuning.
>
> **Q4:**
> "the representation ability of diffusion models does not originate from their generative power." - I don't see where this is investigsted in the proposed manuscript. E.g. the generative power of the proposed model is not shown.
>
> We did not show the generative power of MDM because MDM is not designed to be a generative model. As shown in Section 3.1, the
> generative power of a denoising diffusion model is based on two prerequisites: (1) The forward process should be Gaussian so that Equation 2 and the whole derivation hold. (2) $q(x_{T})$ should be close to $N(0, I)$ so that we can sample from a standard Gaussian distribution and iteratively apply Equation 2 to generate images. MDM apparently does not fulfill these prerequisites (prerequisite 1 does not hold since MDM does not use Gaussian noise; prerequisite 2 does not hold since $q(x_{T})$ in MDM is not $N(0, I)$) and therefore should not be expected to have a good generative capability. Despite that, as discussed in Q1, MDM shows robust representation learning ability. That's why we claim that the representation ability of diffusion models does not originate from their generative power.
>
> **Q5:**
> Will trained model weights also be released on acceptance?
>
> Yes, we will release our code with checkpoints on acceptance.

---

> > ### Author Response · Authors · 2023-11-20
> > **Response to Reviewer caGX Part 2**
> >
> > **Q6:**
> > I am concerned by the apparent lack of a validation set. There seems to be a number of hyper parameters involved such as the choice of timesteps, blocks, patch sizes and training schedules covered in section B. Other choices such as loss function (SSIM or MSE) were also made. It is unclear if test sets were involved in these choices and if so can we expect the reported results to generalize?
> >
> > Thanks for the comment.
> >
> > We did not use a validation set in our paper for two main reasons: (1) We follow the configuration of GlaS and MoNuSeg in [6] and the configuration of FFHQ-34 and CelebA-19 in [1] for a fair comparison. (2) We evaluate the label-efficient attribute of our method in a few-shot setting where the training data is limited and valuable.
> >
> > However, the reported results can still generalize for the following reasons: (1) We use the same data split setting to train and evaluate other comparison methods and do early stopping only based on training loss. Therefore, it is still a fair comparison. (2) We adopt an unsupervised clustering algorithm to choose the blocks (see Figure 6), which does not use any label information in test sets. (3) We intentionally do not tune the hyper parameters for MoNuSeg and CelebA-19, and MDM still achieves state-of-the-art performance. (4) As shown in Table 6 (timesteps) and Table 7 (patch size), even with sub-optimal hyper parameter settings, MDM can still achieve state-of-the-art performance. This implies that MDM is robust to the hyper parameters. (5) For all experiments, we run 10 times with 10 random seeds to make sure that the results are reliable.
> >
> > [1] D. Baranchuk, A. Voynov, I. Rubachev, V. Khrulkov, and A. Babenko. Label-efficient semantic segmentation with diffusion models. In The Tenth International Conference on Learning Representations, ICLR 2022, Virtual Event, April 25-29, 2022. OpenReview.net, 2022.
> >
> > [2] C. Wei, K. Mangalam, P.-Y. Huang, Y. Li, H. Fan, H. Xu, H. Wang, C. Xie, A. Yuille, and C. Feichtenhofer. Diffusion models as masked autoencoder. In ICCV, 2023.
> >
> > [3] W. Xiang, H. Yang, D. Huang, and Y. Wang. Denoising diffusion autoencoders are unified selfsupervised learners. In Proceedings of the IEEE/CVF International Conference on Computer Vision, 2023.
> >
> > [4] W. Tan, S. Chen, and B. Yan. Diffss: Diffusion model for few-shot semantic segmentation. CoRR, abs/2307.00773, 2023.
> >
> > [5] J. Xu, S. Liu, A. Vahdat, W. Byeon, X. Wang, and S. D. Mello. Open-vocabulary panoptic segmentation with text-to-image diffusion models. In IEEE/CVF Conference on Computer Vision and Pattern Recognition, CVPR 2023, Vancouver, BC, Canada, June 17-24, 2023, pages 2955–2966. IEEE, 2023.
> >
> > [6] H. Wang, P. Cao, J. Wang, and O. R. Za ̈ıane. Uctransnet: Rethinking the skip connections in u-net from a channel-wise perspective with transformer. In Thirty-Sixth AAAI Conference on Artificial Intelligence, AAAI 2022, Thirty-Fourth Conference on Innovative Applications of Artificial Intelligence, IAAI 2022, The Twelveth Symposium on Educational Advances in Artificial Intelligence, EAAI 2022 Virtual Event, February 22 - March 1, 2022, pages 2441– 2449. AAAI Press, 2022.

---

> > ### Comment · Reviewer_caGX · 2023-12-04
> > **Satisfied with response**
> >
> > I am satisfied with the response to my concerns. I am not changing the score as I still view it as reasonable given other reviewer and rebuttal comments. All in all it appears to me to be a reasonable experiment with some good results that could be of interest to the community.

---

### Official Review · Reviewer_Sbaz · 2023-11-01

**Soundness:** 2 fair
**Presentation:** 2 fair
**Contribution:** 2 fair
**Rating:** 5
**Confidence:** 3

**Summary:**

This paper presents a novel approach to self-supervised learning using diffusion models. The authors introduce "masked diffusion", where portions of data are probabilistically masked and subsequently reconstructed through a diffusion process. This dynamic masking strategy offers a distinct advantage over traditional static methods, enabling the capture of complex data patterns.

Empirical results showcase the method's excellence, outperforming established self-supervised benchmarks and achieving state-of-the-art performance on multiple datasets.

In essence, this work suggests a potential paradigm shift in self-supervised learning, emphasizing the significance of dynamic masking and diffusion models in data reconstruction and representation.

**Strengths:**

Strengths:

1. **Novel Concept**: The paper presents a new approach in self-supervised learning using diffusion models. The probabilistic mask for data occlusion offers an alternative to traditional static methods, suggesting a different way to approach representation learning.

2. **Empirical Evidence**: The results and ablation studies provide evidence of the method's performance. The proposed technique shows improvements over vanilla MAE, DDPM, and certain traditional models on segmentation datasets.

**Weaknesses:**

Areas of Improvement for the Paper:

1. **Benchmarking for Segmentation Tasks**: The primary focus on segmentation necessitates benchmarking against specialized self-supervised learning (SSL) methods designed for this task, both at the instance-level and pixel/patch-level. A direct comparison with methods such as Leopart, IIC, MaskContrast, DenseCL, MoCoV2, and DINO on standard datasets like COCO and PVOC would provide a holistic evaluation. Refer to paper: Self-Supervised Learning of Object Parts for Semantic Segmentation for methods specific to semantic segmentation.

2. **General SSL Representation Benchmarking**: If general SSL representation learning is the core objective, the method should be contrasted with prevailing SSL techniques (e.g., DINO, MoCoV2) across a spectrum of downstream tasks, including classification, object detection, and linear probing.

3.**Demonstration of Versatility**: To set this work apart, the authors can also consider to showcase results across diverse data domains. Presenting outcomes on datasets related to audio, text, or time-series would emphasize the method's adaptability.

4. **Robustness Against Noisy Data**: The paper could benefit from an evaluation of the model's robustness against noisy or corrupted data, providing a measure of its real-world applicability.

5. **Interpretability and Visualization**: Including a section on interpretability, with visualizations illustrating the diffusion process or the dynamic masking strategy, would help readers better grasp the underlying mechanisms.

6. **Discussion on Limitations**: A more explicit section discussing the method's limitations, potential pitfalls, or scenarios where it might not be the best fit would offer a balanced perspective.

**Questions:**

If some points in the weaknesses section are addressed, the score can be increased.

---

> ### Author Response · Authors · 2023-11-20
> **Response to Reviewer Sbaz**
>
> Thank you for your careful reading of our paper and your valuable comments, which
> have helped us to improve the presentation of our manuscript. Please find our answers to your
> remarkable comments in the following:
>
> **Q1:**
> Benchmarking for Segmentation Tasks.
>
> We totally agree that these experiments are important and meaningful. In our manuscript, we focus on the semantic segmentation under a label-efficient setting. We plan to conduct a direct comparison with methods such as Leopart on COCO and PVOC in our future work.
>
> **Q2:**
> General SSL Representation Benchmarking.
>
> Thanks so much for the constructive suggestion.
>
> In the revised manuscript, we constrain our objective to representation learning specifically for segmentation, rather than general SSL representation learning.
>
> Nevertheless, we evaluate our method on other downstream tasks and get promising results compared to other prevailing SSL techniques.
> 		In particular, we conduct the MDM pre-training on CIFAR-10 and then use linear probing to test how well MDM performs on classification task. The experimental details and results are shown in Appendix C. We provide the results below:
> | Learning Type      | Method   | Network    | Accuracy (%) |
> |------------------- |----------|------------|--------------|
> | Contrastive        | MoCov1   | ResNet-50  | 85.0         |
> | Contrastive        | MoCov2   | ResNet-50  | 93.4         |
> | Contrastive        | SimCLR   | ResNet-50  | 94.0         |
> | Contrastive        | SimCLRv2 | ResNet-101 | 96.4         |
> |                   |          |            |              |
> | Generative         | DDPM     | U-Net      | 90.1         |
> | Generative         | MDM      | U-Net      | 94.8         |
>
> The table above shows that MDM achieves much better linear evaluation accuracy than DDPM which has the same learning type and network and competitive performance compared to the contrastive learning methods.
>
> **Q3:**
> Visualization of the dynamic masking strategy.
>
> In the revised manuscript, we have added the visualization of the dynamic masking strategy in Figure 1.
>
> **Q4:**
> Demonstration of Versatility.
>
>  We totally agree that adapting the method to other data domains can be interesting. However, many extra modifications need to be made due to the domain gap. We think it will be an interesting and meaningful direction of further investigation.
>
> **Q5:**
> Robustness Against Noisy Data to Measure The Method's Real-world Applicability.
>
> Thanks for giving this constructive suggestion. We agree with the reviewer that a robustness analysis is helpful.
>
> We added robustness analysis in Appendix B. We apply 15 types of corruption from [1] to the test data
>  (Gaussian Noise, Shot Noise, Impulse Noise, Defocus Blur, Frosted Glass Blur, Frost, Brightness, Contrast, Elastic, Pixelate, JPEG, Speckle, Gaussian Blur, Spatter and Saturat). As shown in Figure 8, MDM demonstrates strong robustness against various types of corruption, even without employing any data augmentation related to the applied corruption (e.g., noise) during both pre-training and downstream segmentation training.
>
> **Q6:**
> Discussion on Limitations.
>
> We added the discussion of the method's limitations and future work in section 6.
>
> [1] D. Hendrycks and T. G. Dietterich. Benchmarking neural network robustness to common corruptions and perturbations. In 7th International Conference on Learning Representations, ICLR 2019, New Orleans, LA, USA, May 6-9, 2019. OpenReview.net, 2019.

---

### Official Review · Reviewer_x7rC · 2023-11-01

**Soundness:** 2 fair
**Presentation:** 3 good
**Contribution:** 3 good
**Rating:** 6
**Confidence:** 3

**Summary:**

The paper proposes a pre-training strategy for image segmentation motivated by the recent success of diffusion models. First, an image is masked with different masking ratios, t. Then, the masked image and the mask ratio are given as input to a Unet to restore the original image with SSIM loss. This architecture is called the Masked Diffusion Model (MDM). Once MDM is trained with unlabeled data, a small segmentation network that takes the representation of a decoder as input is trained with a supervised loss on the available labeled data.
The experiments are presented on both medical and natural image datasets. The results demonstrate that MDM achieves better performance compared to some SoTA self-supervised learning methods such as MAE and DDPM.

**Strengths:**

- The paper present an extensive experimental evaluations on 2 natural and 2 medical image data sets with ablation studies.

**Weaknesses:**

- My main concern is the novelty of the method. The paper mentions that with the fixed t, the method degrades to a vanilla masked autoencoder with SSIM loss. This basically means that the only contribution of the paper is masking the image with a dynamic masking ratio during training, which concerns me regarding the contribution of the paper.

- Although the improvement achieved by this small change is interesting on Glas 10% case (IOU is 76.19 for MAE and 82.70 MDM with MSE; which is quite a significant improvement with only this small change (assuming MAE in table 1 is trained with MSE loss)), the results on FFHQ-34 shows that this finding cannot be generalized across datasets (IoU of MAE is 57.06 while MDM with MSE is 55.06).

- Table 4 shows that using SSIM improves the performance significantly both for DDPM and MDM while MDM is still being better. I think a more fair comparison of MDM w/ SSIM loss would have been with MAE w/ SSIM loss, since it would show the effect of changing t dynamically.

- It is not very clear from the paper which decoder level representation is given as input to the segmentation network. Is it only the last layer or some combinations of the last few layers? I also think it would be interesting to know the performance change as a function of using different decoder level features. Additionally, for fair comparison, the same level decoder features should be used for also DDPM and MAE. I didn't see a clear statement about this in the paper.

**Questions:**

- Please address my concerns in the weaknesses section, especially the ones related to the contribution.

---

> ### Author Response · Authors · 2023-11-20
> **Response to Reviewer x7rC**
>
> We would like to thank you for reviewing our paper and providing valuable and helpful comments. Please find our responses to your comments in what follows:
>
> **Q1:**
> The contribution of this paper.
>
> The reviewer proposes that the contribution is limited because the technical implementation is simple. However, we respectfully disagree with this assessment as Reviewer Sbaz and Reviewer caGX agree unanimously that the proposed method is indeed novel. From the self-supervised learning perspective, even the famous denoising diffusion probability models (DDPM) can also be treated as denoising noisy image with different levels of noise during training, and the technical implementation is really simple as well. The simplicity of implementation does not diminish the effectiveness of MDM as a robust self-supervised representation learning method, much like DDPM's simplicity doesn't detract from its prowess as an impressive image synthesizer.
>  To alleviate the reviewer's concern, we summarize our contributions below:
>
>    - In this paper, we analyze the diffusion models from the self-supervised learning perspective and extend denoising diffusion models into a self-supervised pre-training algorithm, using denoising as a preliminary task. We then empirically verify that the representation ability of diffusion models does not originate from their generative power by proposing our MDM which does not have generative power but still possesses powerful representation learning ability. This conclusion is significant since it allows researchers not to focus too much on the generative performance of diffusion models if their goal is to acquire meaningful representations. In practice, many works [1, 2, 3] which use extracted diffusion representations for downstream segmentation tasks can be potentially improved.
>    - Our MDM offers a different way to approach representation learning. We conduct extensive experiments to validate the effectiveness of our method. The state-of-the-art segmentation performance and competitive classification performance show the superiority of the proposed MDM.
>    - The label-efficient attribute of MDM has a wide range of applications. In particular, when developing medical AI, it is usually easy to collect large number of images but not feasible to make the corresponding labels for all the images. The proposed method can be a perfect solution for such scenarios.
>
> **Q2:**
> The fair comparison with MAE should be added and the results on FFHQ-34 implies the improvement brought by MDM cannot generalize.
>
> We appreciate this constructive suggestion. We agree that MDM w/ SSIM loss should be compared with MAE w/ SSIM loss for a more fair comparison. We updated our new results in Table 4:
>
> | Method | Loss Type | GlaS 10% (8) | FFHQ-34 |
> |--------|-----------|--------------|---------|
> | DDPM | MSE | 82.32±0.77 | 58.75±0.16 |
> | | | | |
> | DDPM* | MSE | 78.77±1.01 | 51.63±0.16 |
> | | **SSIM** | **81.79±1.13** | **56.97±0.18** |
> | | | | |
> | MAE | **MSE** | **79.47±1.08** | **57.06±0.20** |
> | | SSIM | 76.72±1.75 | 56.55±0.13 |
> | | | | |
> | MDM | MSE | 82.70±0.79 | 55.06±0.21 |
> | | **SSIM** | **84.51±1.15** | **59.18±0.11** |
>
> We can draw two conclusions from this table:
>
>    - SSIM loss is not suitable for MAE. From our empirical results, SSIM loss can encourage the model to learn the structure information of the whole image because it takes into account structure while MSE loss assumes pixel-wise independence, which is beneficial for downstream dense prediction tasks. However, MAE calculates the loss per patch and only on invisible parts. Using SSIM loss to substitute MSE loss therefore does not make sense in MAE. Similarly, it is also not reasonable to use SSIM loss for DDPM when its reconstruction objective is the added noise.
>
>   - When adopting each corresponding suitable loss function, MDM achieves better performance than MAE and DDPM on both GlaS and FFHQ-34.
>
> **Q3:**
> Questions regarding the layers of decoder representation.
>
>  We are sorry not making this information clear enough.
>
>   In fact, we discussed the block choosing strategy in Appendix B of our original paper. Also, we visualized the clustering of features extracted from different levels of the decoder for MDM and DDPM in Figure 6. The features from the deeper layers (blocks with smaller values) exhibit coarse semantic masks, while those from shallower layers (blocks with larger values) reveal finer details yet lack the same level of semantic coherence for coarse segmentation. Therefore, we choose the middle blocks B = \{8, 9, 10, 11, 12\} and B = \{5, 6, 7, 8, 9, 10, 11, 12\} among the 18 decoder blocks for two medical image datasets and two natural image datasets, respectively.
>
>   For a fair comparison, MDM and DDPM use the same blocks for feature extraction. Due to the fact that MAE uses a different architecture (ViT) from MDM and DDPM (U-Net), we extract feature maps from the deepest 12 ViT-L blocks for MAE, following [1].

---

> > ### Author Response · Authors · 2023-11-20
> > **References**
> >
> > [1] D. Baranchuk, A. Voynov, I. Rubachev, V. Khrulkov, and A. Babenko. Label-efficient semantic segmentation with diffusion models. In The Tenth International Conference on Learning Representations, ICLR 2022, Virtual Event, April 25-29, 2022. OpenReview.net, 2022.
> >
> > [2] W. Tan, S. Chen, and B. Yan. Diffss: Diffusion model for few-shot semantic segmentation. CoRR, abs/2307.00773, 2023.
> >
> > [3] J. Xu, S. Liu, A. Vahdat, W. Byeon, X. Wang, and S. D. Mello. Open-vocabulary panoptic segmentation with text-to-image diffusion models. In IEEE/CVF Conference on Computer Vision and Pattern Recognition, CVPR 2023, Vancouver, BC, Canada, June 17-24, 2023, pages 2955–2966. IEEE, 2023.

---

> > ### Comment · Reviewer_x7rC · 2023-11-21
> > **Further comments**
> >
> > Thanks to the authors for addressing my comments.
> >
> > I am still not very convinced by the experiments where MAE is trained with SSIM. I understand SSIM is unsuitable for the vanilla MAE because the loss is only computed over the masked patches. However, the SSIM loss for MAE can be computed in a manner similar to that employed for the MDM method by fixing t in MDM.
> >
> > Such an experiment already exists in the paper for the Glas dataset in Table 3. I assume the setting "MDM with fixed t" corresponds to MAE with SSIM loss. However, I have concerns regarding this experiment:
> >
> > - T is fixed to 50 for MDM. This means that MAE with SSIM is only tested at T=50. However, we do not know that T=50 is the best hyperparameters for MAE with SSIM. I doubt this because  MAE with SSIM with T=50 is even lower than MAE with MSE (e.g. IoU 79.67 vs 81.35 in Glas 100%). If this is the case, then there is the question of why SSIM contributes more to MDM than MAE. The authors' answer to this question is, "However, MAE calculates the loss per patch and only on invisible parts. Using SSIM loss to substitute MSE loss therefore does not make sense in MAE.", however, this is not the case anymore since SSIM for should be computed as in the MDM experiments.
> >
> > - The above question can be answered with empirical results (by carefully tuning T). For this, we need results on multiple datasets such as FFHQ and CelebA. Currently, there are only results for the Glas dataset in Table 3.

---

> > > ### Author Response · Authors · 2023-11-21
> > > **Further responses**
> > >
> > > Thanks for your further comments. We address your concerns below:
> > >
> > > MAE is not tested at T=50. During pre-training, we follow the official setting of MAE [1], where 75\% of the image is masked (equivalently, in our experiment, T=750). During fine-tuning, the whole image is fed to the MAE as input, following the official setting. According to the results and discussion in [1], a 75\% masking ratio works best for MAE (see Figure 5 in [1]). Therefore, it is noteworthy that all MAE results in Table 2, Table 3, and Table 4 are acquired under the best setting for MAE.
> > >
> > > Regarding the concern that MAE with SSIM at T=50 is even lower than MAE with MSE (e.g., IoU 79.67 vs. 81.35 in Glas 100\%):
> > >
> > > As discussed above, the true comparison acquires the results, i.e., IoU 79.67 vs. 81.35 in Glas 100\%, as follows: MDM with a fixed T=50 using SSIM loss and MAE with a 75\% masking ratio using MSE loss (the best setting for MAE). The fact that the result of an MDM with a fixed t is slightly lower than that of MAE is reasonable for two reasons: (1) MAE uses a ViT architecture while our MDM uses U-Net; (2) According to the MAE paper [1], a 75\% masking ratio is the best, not the T=50 (roughly 5\% masking ratio) used for our MDM with a fixed t. We use a fixed t=50 for MDM in order to fairly compare with the proposed MDM since it uses t=50 for feature extraction.
> > >
> > > We further explain all the results to make them clear:
> > >
> > > 1. For Table 1 and Table 2, all the methods, including MAE, use their corresponding best settings to achieve the best results. MDM achieves the best performance among them.
> > >
> > > 2. For Table 3, we introduce the results of MAE only for an intuitive presentation of how SSIM loss helps our MDM. MDM with a fixed t cannot be simply treated as MAE for the reasons discussed above.
> > >
> > > 3. For Table 4, we use the best setting for each method and evaluate both SSIM loss and MSE loss.
> > >
> > > [1] K. He, X. Chen, S. Xie, Y. Li, P. Doll ́ar, and R. B. Girshick. Masked autoencoders are scalable
> > > vision learners. In IEEE/CVF Conference on Computer Vision and Pattern Recognition, CVPR
> > > 2022, New Orleans, LA, USA, June 18-24, 2022, pages 15979–15988. IEEE, 2022.8

---

> > > > ### Comment · Reviewer_x7rC · 2023-11-21
> > > >
> > > > Thanks for the response and clarification.
> > > >
> > > > - I apologize for my confusion. I overlooked that the values IoU 79.67 vs 81.35 in Glas 100% are not directly comparable because the architecture, loss, and masking ratios are different in both experiments. These results do not show the effect of SSIM loss.
> > > >
> > > > - Let me clarify my previous comments. I would like to see the answers to the following questions in the paper:
> > > >
> > > > Q1) How does MAE perform if it is trained with SSIM loss?
> > > > - This can be shown in 2 ways:
> > > > 1.1. Training the original MAE with ViT architecture with SSIM loss by applying the loss to all patches as in MDM. The table in the authors' first response was obtained with SSIM loss applied on the masked patches only, which is unsuitable, as also stated by the authors.
> > > > 1.2. Performing experiments by fixing t in MDM. This setting corresponds to MAE with a UNet instead of ViT. However, t should be selected carefully based on a validation set accuracy. Such an experiment already exists in the paper in Table 3 for the Glas dataset with t=50. This corresponds to roughly a 5% masking ratio which is significantly lower than the suggested (75%) in the original MAE paper. I understand that setting t using a validation set might require some time, but then the ration should be set to 75% as suggested in the MAE paper.
> > > >
> > > > Q2) Does the result in Q1 hold for other datasets?
> > > > The experiments to answer Q1 should be done for multiple dataset to show the results generalizes to multiple datasets.
> > > >
> > > > - I have one additional experiment that might be useful for the future could be MDM with the ViT architecture as in the MAE paper.

---

> > > > > ### Author Response · Authors · 2023-11-23
> > > > > **Response to Reviewer x7rC**
> > > > >
> > > > > **Q1:**
> > > > > How does MAE perform if it is trained with SSIM loss? Does the
> > > > > result hold for other datasets?
> > > > >
> > > > > Thanks for your recommended 2 ways to conduct the extra experiments. 1.1 is not available during the left rebuttal period because MAE pre-training takes more than 4 days using our GPUs. Therefore, we conduct the experiments in 1.2 to alleviate your concern:
> > > > >
> > > > > - We fix t=750 (masking ratio 75%) as suggested in MAE and train MDM using SSIM loss on Glas and MoNuSeg. This setting corresponds to MAE with a UNet instead of ViT (still slightly different). The results are provided below:
> > > > >
> > > > > | Method               | GlaS 100% (85) || GlaS 10% (8) || MoNuSeg 100% (30) || MoNuSeg 10% (3) ||
> > > > > |----------------------|----------|---------|----------|---------|----------|---------|----------|---------|
> > > > > |                      | Dice (%) | IoU (%) | Dice (%) | IoU (%) | Dice (%) | AJI (%) | Dice (%) | AJI (%) |
> > > > > | MDM fixed t = 50     | 88.68$\pm$0.54 | 79.67$\pm$0.86 | 86.82$\pm$1.04 | 76.72$\pm$1.59 | - | - | - | - |
> > > > > | MDM fixed t = 750    | 90.37$\pm$0.84 | 82.43$\pm$1.39 | 86.95$\pm$2.42 | 76.99$\pm$3.70 | 77.63$\pm$0.45 | 63.61$\pm$0.57 | 77.03$\pm$0.53 | 62.83$\pm$0.68 |
> > > > > | **MDM**              | **91.95$\pm$1.25** | **85.13$\pm$2.09** | **91.60$\pm$0.69** | **84.51$\pm$1.15** | **81.01$\pm$0.35** | **68.25$\pm$0.49** | **79.71$\pm$0.75** | **66.43$\pm$1.02** |
> > > > >
> > > > > The results are slightly better than MDM with t = 50 but still worse than our MDM. These results
> > > > > again verify the effectiveness of our MDM.
> > > > >
> > > > > **Q2:**
> > > > > I have one additional experiment that might be useful for the future
> > > > > could be MDM with the ViT architecture as in the MAE paper.
> > > > >
> > > > > - Thanks for the suggestion. We have updated our future work in Section 6 that more architectures including ViT can be explored in the future.

---

> > > > > > ### Comment · Reviewer_x7rC · 2023-11-23
> > > > > >
> > > > > > Thanks to the authors for the additional experiments. Now, we can see that MAE w/ UNet+SSIM outperforms MAE w/ ViT+MSE despite using a more lightweight architecture, even without a carefully tuned t for the former.
> > > > > >
> > > > > > MDM still benefits from using dynamic t since t is a crucial parameter and it eliminates the need for fine-tuning this parameter. I think this is quite useful, and I now view the paper more positively. I will update my score accordingly.

---

### Official Review · Reviewer_V2xK · 2023-11-03

**Soundness:** 2 fair
**Presentation:** 2 fair
**Contribution:** 2 fair
**Rating:** 3
**Confidence:** 4

**Summary:**

This paper applies masking to diffusion models and shows improvement in segmentation tasks for both medical and natural images. The authors also try to use SSIM instead of MSE for pre-training in order to improve the downstream segmentation performance.

**Strengths:**

1. The statistical results shown in Table 1 look promising if all the methods are compared fairly.

**Weaknesses:**

1. The writing of this paper needs to be improved. Many claims in the introduction section are not very well-supported (e.g. "such efforts risk deviating from the theoretical underpinnings of diffusions") and are not very well organized.
2. It is not convincing enough to conclude that the representation learned is better while only tested on segmentation downstream tasks.
3. The choice of SSIM over MSE is rather empirical and not well justified.

**Questions:**

1. The authors chose SSIM over MSE to improve the segmentation performance. However, there are also other types of choices such as normalized cross-correlation, MAE etc. It is also more of a trade-off between learning a general representation than tuning toward specific downstream tasks (e.g. segmentation in this case). The authors need to justify more about this.

2. Since the goal of the proposed method is to learn a better representation, how well the method performs on other downstream tasks?

3. Figure 2 and Figure 3 are quite repetitive and it is very hard to conclude which methods are better. Reporting Dice scores along each figure will be helpful.

---

> ### Author Response · Authors · 2023-11-20
> **Response to Reviewer V2xK**
>
> We would like to thank you for reviewing our paper and providing valuable and helpful comments. Please find our responses to your comments in what follows:
>
> **Q1:**
> The statistical results shown in Table 1 look promising if all the methods are compared fairly.
>
> - Thanks for the positive comment. We tried our best to make sure all the experiments being fair:
>   - We use the same settings for all methods unless it is infeasible.
>   - We use each method's official implementation and checkpoints (if released).
>   - We conduct all the experiments 10 times using 10 random seeds and report the standard deviation and mean.
>
> **Q2:**
> The writing of this paper needs to be improved.
>
> - We have revised the manuscript for most parts which may cause confusion and reorganized the manuscript. The new manuscript is much clearer and the claims are well-supported.
> Regarding the specific claim you mentioned, we say that ''such efforts risk deviating from the theoretical underpinnings of diffusion'' for the reason that: As shown in Section 3.1, the whole theoretical derivation of denoising diffusion model is based on the assumption that the tractable posterior used to generate images from a standard Gaussian distribution is Gaussian, which is fulfilled in denoising diffusion by adding Gaussian noise in the forward process. However, recent methods such as Cold Diffusion [1] have replaced the Gaussian noise with other corruptions which can not be described by Gaussian distributions, making them deviate from the theoretical underpinnings of diffusion and suffer from worse sample quality.
>
> **Q3:**
> How well the method performs on other downstream tasks?
>
> - To evaluate the performance of our method on other downstream tasks, we conduct the MDM pre-training on CIFAR-10 and then use linear probing to test how well MDM performs on a classification task. The experimental details and results are shown in Section C of the Appendix. We provide the results below:
>
> | Learning Type      | Method   | Network    | Accuracy (%) |
> |------------------- |----------|------------|--------------|
> | Contrastive        | MoCov1   | ResNet-50  | 85.0         |
> | Contrastive        | MoCov2   | ResNet-50  | 93.4         |
> | Contrastive        | SimCLR   | ResNet-50  | 94.0         |
> | Contrastive        | SimCLRv2 | ResNet-101 | 96.4         |
> |                   |          |            |              |
> | Generative         | DDPM     | U-Net      | 90.1         |
> | Generative         | MDM      | U-Net      | 94.8         |
>
>
> &ensp; &ensp; The table above shows that MDM achieves much better linear evaluation accuracy than DDPM which has the same learning type and network and competitive performance compared to the contrastive learning methods.
>
> **Q4:**
> Justify the choice of SSIM over MSE. Is it a trade-off between learning a general representation than tuning toward specific downstream tasks?
>
> - From the experimental results, we observe a phenomenon that even the MSE loss is quite low during pre-training (which indicates that the image restoration is good based on MSE loss), the segmentation performance of the downstream task using the learned representations is not always good. The reason could be that MSE loss assumes pixel-wise independence. As another commonly used loss function in image restoration task, SSIM loss can encourage the model to learn the structure information of the whole image because it takes into account structure, which is beneficial for downstream dense prediction task.
>
>     Regarding the trade-off between learning a general representation than tuning toward specific downstream tasks, we believe that better loss functions than SSIM loss must exist for specific datasets and downstream tasks. We do not intend to claim SSIM loss is only choice.
>
>     However, we do not think using SSIM loss in our MDM is a trade-off for the reason that our MDM with SSIM loss not only achieves state-of-the-art semantic segmentation performance on all four datasets we used, but also demonstrates competitive classification performance on CIFAR-10, as discussed above.
>
> **Q5:**
> Figure 2 and Figure 3 are quite repetitive and it is very hard to conclude which methods are better. Reporting Dice scores along each figure will be helpful.
>
> - Thanks for pointing this out. It is indeed hard to conclude which methods are better from Figure 2 and Figure 3 in the original manuscript. In the revised manuscript, we report the dice score of each prediction in the corner for Figure 3 and Figure 4 (Figure 2 and Figure 3 in previous version).
>
> [1] A. Bansal, E. Borgnia, H.-M. Chu, J. S. Li, H. Kazemi, F. Huang, M. Goldblum, J. Geiping, and T. Goldstein. Cold diffusion: Inverting arbitrary image transforms without noise, 2022.

---

### Author Response · Authors · 2023-11-20
**Response to the reviewers**

We thank the reviewers for the critical assessment of our work and constructive feedback.

Before we reply to them individually and address their concerns, we report some common concerns:

1. **Whether our proposed representation learning scheme is specific to segmentation tasks.**
   - This is indeed a very important question and also not well explained in our manuscript. In this revision, we made two major changes:
      - Adjusting our claim on the contribution, adding discussion on limitation and future works in terms of versatility of the learned representation.
      - Adding experimental results on other downstream tasks beyond segmentation, such as linear probing on CIFAR-10. Our results on CIFAR-10 show comparable improvement as the segmentation tasks. In short, in our revised version, we show similar success on another downstream task beyond segmentation, meantime avoiding any potential overclaim on the versatility of the learned representation.

2. **The choice of SSIM loss needs to be justified and MAE w/ SSIM loss should be compared.**
   - We added the experiments of MAE w/ SSIM loss on GlaS and FFHQ-34 for a fair comparison with our method in Table 4. Detailed justification and discussion are provided in the responses to Reviewer V2xK and Reviewer x7rC.

3. **The presentation can be further improved.**
   - We have updated some figures and revised the manuscript for better clarity.

---

> ### Author Response · Authors · 2023-11-23
> **Looking forward to hearing from you**
>
> Dear Reviewers,
>
> As the deadline of ICLR rebuttal period is approaching, we look forward to hearing your feedback on our responses. We would be happy to address any remaining concerns that you may still have.
>
> Thanks,
>
> Authors

---

### Meta-Review · Area_Chair_Th8W · 2023-12-14

**Metareview:**

This paper explores the relationship between generative performance and representation learning in diffusion models. It introduces the Masked Diffusion Model (MDM), which uses a masking mechanism instead of traditional additive Gaussian noise. During the review process, the reviewers raised concerns about the specificity of the proposed method to segmentation tasks, the choice of SSIM loss, the presentation quality, and the performance on other downstream tasks. The authors responded to these concerns by adjusting their claims, adding new experimental results, discussing the choice of SSIM loss, and improving the manuscript's presentation. Reviewer x7rC, initially giving a score of 3, raised their score to 6 after the authors' rebuttal addressed their concerns. However, Reviewer V2xK did not participate in the rebuttal phase (but provided updated comments recently after AC reached out, who showed reluctance to further increase the score). Reviewer Sbaz maintained a score of 5 after the rebuttal. Overall, AC decided to recommend a reject decision and encourage the authors to further improve their manuscript based on the reviewers' feedback.

==
For visibility -- V2xK's additional comment to AC:

"I have read again the paper and the authors' reply to my initial comments but I am still not super convinced to shift my opinion towards acceptance, mainly because 1) the evaluation feels not strong enough even with the additional classification task the authors added for the rebuttal (only two tasks and MDM is just comparable to contrastive type model for classification) 2) the choice of SSIM over MSE still seems rather empirical than thoughtful like it just happens to work out for this case instead of having a strong analysis to explain the rationale (like the authors admit that SSIM is not the only choice in the rebuttal)."

**Justification For Why Not Higher Score:**

The paper has great potential and the direction is a really important one. However, some concerns including the narrow focus in evaluation, more convincing justification of the SSIM loss, and presentation/clarity Issues can be further improved/addressed.

**Justification For Why Not Lower Score:**

N/A

---

### Decision · Program_Chairs · 2024-01-16

Reject